# Southern-style *Pad Thai* sauce: From traditional culinary treat to convenience food in retortable pouches

**Yanyong Cheenkaew, Worawan Panpipat, Manat Chaijan** *

Department of Agro-Industry, Food Technology and Innovation Research Center of Excellence, School of Agricultural Technology, Walailak University, Nakhon Si Thammarat, Thailand

* cmanat@wu.ac.th

**Data Availability Statement:** All relevant data are within the manuscript and its Supporting Information files.

**Funding:** This study was funded by the Thailand Research Fund (TRF) under the Research and

## Abstract

*Pad Thai*, a Thai dish of stir-fried rice noodles and other ingredients, is one of the culinary heritages of Thailand. In the southern region of Thailand, *Pad Thai* has different characteristics from other areas because coconut milk and curry paste are used to produce a thick sauce prior to cooking. To commercially distribute this uniquely local culinary treat in a convenient form all over Thailand and other export markets, a shelf-stable sauce using heat sterilization should be developed. Retort processing technology with retort pouches can be used for this purpose. However, phase separation and lipid oxidation can occur and subsequently reduce the overall quality of retorted southern-style *Pad Thai* sauce. The application of an appropriate stabilizer and antioxidant can be used to prevent such problems. Thus, the objective of this study was to investigate the effects of stabilizers and antioxidants on the stability of retorted southern-style *Pad Thai* sauce. Southern-style *Pad Thai* sauce was prepared according to the traditional recipe in the presence of different stabilizers (2.3% potato starch (PS) + 0.1% xanthan gum (XG), 0.5% soy lecithin (LT), and 4% whey protein isolate (WPI)) and antioxidants (500 mg/kg ascorbyl palmitate + 500 mg/kg α-tocopherol (As + Toc), 100 mg/kg ethylenediaminetetraacetic acid (EDTA), 200 mg/kg butylated hydroxytoluene (BHT), and As+Toc+EDTA+BHT (mixed antioxidants)). Samples were packed in retort pouches and processed in a retort at 121˚C with an $F_0$ value of 3.57 min. Results showed that the retorted southern-style *Pad Thai* sauce can be stabilized by 2.3% PS plus 0.1% XG in combination with mixed antioxidants. PS and XG helped stabilize the sauce with a desirable viscosity, water holding capacity, and color without any creaming layer and negative effect on sensory properties. Mixed antioxidants improved the oxidative stability of the retorted sauce by retarding the changes in the peroxide value and color during retorting. Consequently, the processing of southern-style *Pad Thai* sauce in retortable pouches could increase the market demand for this traditional product due to its convenience and ready-to-use features.

## Introduction

Traditional foods represent the cultural heritage of a particular region. The ingredients and preparation methods of traditional foods can vary considerably depending on the cultural

Researcher for Industries (RRI) Program (Grant No. MSD61I0038). The funders had no role in study design, data collection and analysis, decision to publish, or preparation of the manuscript.

**Competing interests:** The authors have declared that no competing interests exist.

richness of a region [1, 2]. *Pad Thai* is a popular Thai dish of stir-fried rice noodles and other seasonings [3]. The style of *Pad Thai* in Southern Thailand differs from other areas because coconut milk and curry paste are used to produce a thick sauce prior to adding rice noodles and other condiments. Especially in the Pak Phanang District of Nakhon Si Thammarat Province in Thailand, coconut milk, nipa palm sugar, salt, shrimp paste, and chili paste are used to produce an emulsion-based sauce leading to the special flavor of this southern-style *Pad Thai*, so-called *Pak Phanang Mee Pad* (spicy stir-fried rice noodles with coconut milk) (Fig 1A). This product symbolizes the Pak Phanang River Basin of Nakhon Si Thammarat, and it is a specialty dish in Southern Thailand which can attract the gastronomic tourism of this area. Accordingly, the *Pak Phanang Mee Pad* fair is organized annually by the Tourism Authority of Thailand.

In order to commercially distribute this uniquely local culinary treat all over Thailand and other export markets, the production of shelf-stable sauce using heat sterilization should be developed. Retort processing technology, one of the thermal processing methods, can be used to achieve this goal. Nowadays, retorted foods in pouches are acceptable worldwide due to their lighter, more appealing, and more convenient end use and longer shelf-life [4]. However, based on the emulsion-based system that produces other emulsified sauces [5, 6, 7, 8, 9], the retort process of southern-style *Pad Thai* sauce may encounter phase separation and lipid oxidation during retorting. It was reported that the retorting of emulsion at high temperatures (121°C for 16 min) resulted in lipid oxidation and an increase in coalescence of oil droplets [5]. Lipid oxidation is a critical issue in regard to emulsified foods because it is considered as an interfacial phenomenon [10, 11]. The large interfacial area of emulsion provides numerous sites for lipid oxidation to occur [10, 11]. Also, the separation of the solid and liquid phase is one of the main problems associated with the quality of sauces [6]. Polysaccharides and protein hydrocolloids are commonly used as stabilizers of food due to their thickening and gelling properties, which can capture and hold moisture, inhibit its evaporation from foodstuffs, stabilize emulsions, and control rheology [7]. The type and concentration of the applied hydrocolloids and their combinations, the use of the final product, and consumer demands influence the efficiency of hydrocolloids in food products [7]. Whey proteins can play important roles as an emulsifier and stabilizer in heat-treated emulsion [8]. Polysaccharides (e.g. xanthan gum and modified starch) were found to significantly improve emulsion stability against creaming during heat treatment [5, 9, 12]. In addition, lipid-based stabilizers (e.g. commercial lecithins) increased the heat stability of emulsions [13]. Generally, oxidation in emulsions can occur via a free radical route, and it can be triggered by permeation of free radicals generated at the emulsion interface [14]. The addition of free radical scavenging antioxidants is one of the practical strategies to prevent lipid oxidation in food emulsions [15]. Butylated hydroxytoluene (BHT) and ethylenediaminetetraacetic acid (EDTA) were authorized as synthetic antioxidants for use in food [16]. Antioxidant synergism between tocopherols and ascorbyl palmitate in food products has been reported by Bruun-Jensena et al. [17] and Cort [18]. Thus, free radical scavengers and other antioxidants can be used to retard oxidation in southern-style *Pad Thai* sauce. Therefore, the objective of this study was to determine the effects of stabilizers and antioxidants on the overall quality of retorted southern-style *Pad Thai* sauce.

## Material and methods

### Food additives and chemicals

Potato starch (PS) was purchased from Continental Food Co., Ltd. (Bangkok, Thailand). Xanthan gum (XG), soy lecithin (LT), whey protein isolate (WPI), EDTA, BHT, α-tocopherol and ascorbyl palmitate were purchased from Chemipan Corporation Co., Ltd. (Bangkok,

Thailand). Other chemicals used for analyses e.g. potassium iodide, sodium thiosulfate, chloroform and acetic acid were obtained from Sigma Aldrich Co. (St. Louis, MO, USA).

## Production of southern-style *Pad Thai* sauce

Southern-style *Pad Thai* sauce was prepared using a traditional Thai process. The ingredients for the sauce consisted of 34.72% water, 17.36% coconut milk, 16.67% sugar, 13.89% nipa palm sugar, 10.42% red onion, 5.03% chili paste, 1.56% shrimp paste and 0.35% salt. All ingredients used in the preparation of the sauce were obtained from a local market in Thasala, Nakhon Si Thammarat. To prepare the sauce, all ingredients were blended and stir-fried in an open pan (90-100˚C) for approximately 12 min (Fig 1B).

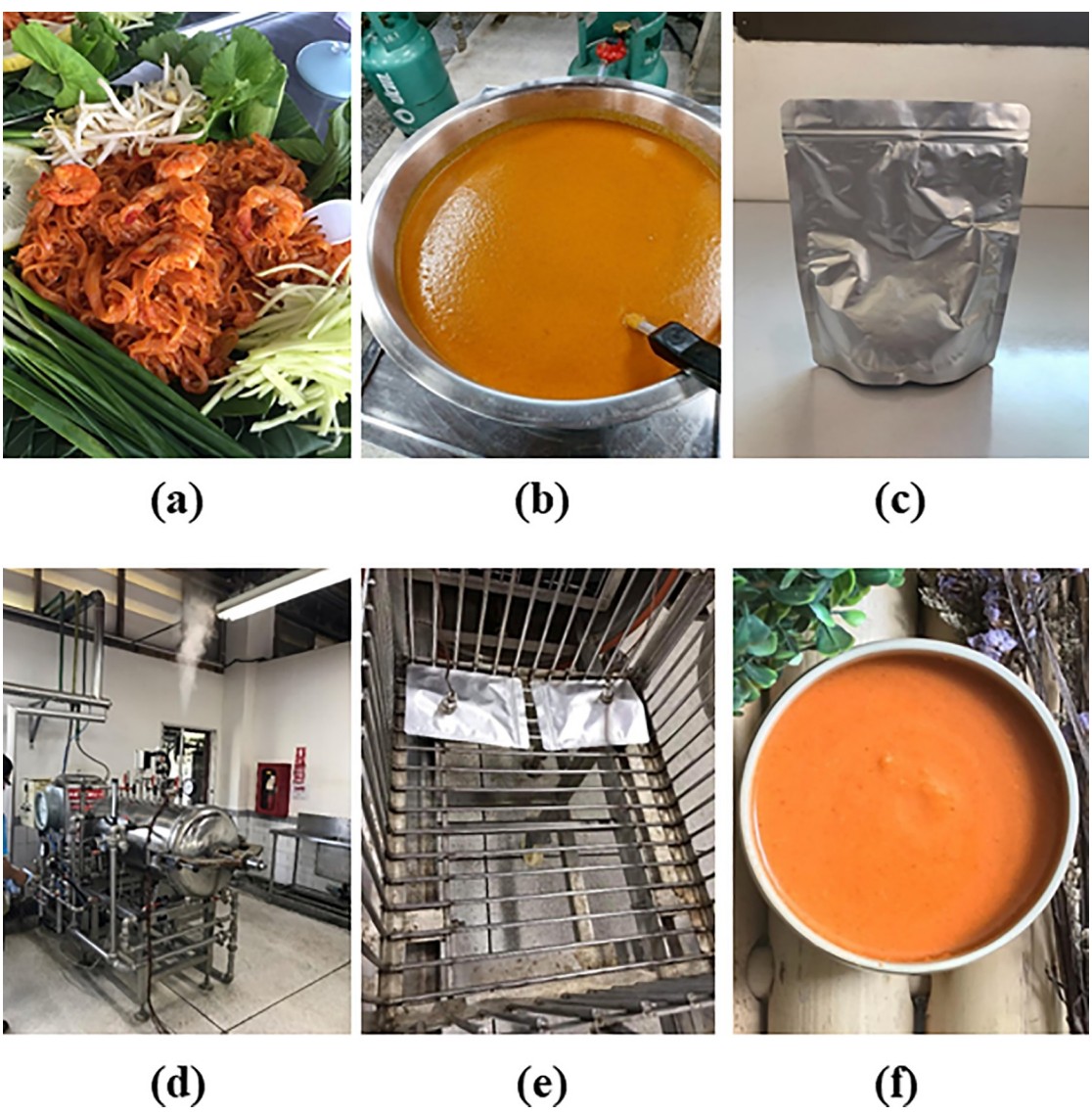

**Fig 1.** Southern-style *Pad Thai*, so-called *Pak Phanang Mee Pad* (spicy stir-fried rice noodles with coconut milk) (a), preparation of southern-style *Pad Thai* sauce (b), retortable pouch used to fill the product (c), pilot-scale horizontal stationary retort system (d), monitoring the temperatures in the retort and pouch center with thermocouples (e), and appearance of the final retorted sauce (f).

## Effects of stabilizers on retorted southern-style *Pad Thai* sauce

To study the effects of stabilizers on retorted southern-style *Pad Thai* sauce, different stabilizers including 2.3% PS plus 0.1% XG, 0.5% LT, and 4% WPI were separately added at the stir-frying step during the preparation of the sauce to obtain four different sauces–namely, the control (without stabilizer), sauce with PS + XG, sauce with LT, and sauce with WPI. The concentrations of PS and XG were previously optimized using response surface methodology (RSM), and the optimal ratio (2.3% PS + 0.1% XG) was used in this study in comparison with a reported lipid-based stabilizer (0.5% LT, [19]) and protein-based stabilizer (4% WPI, [8]).

For the retort process for sterilization, pre-fabricated multilayer laminated retortable pouches (12 μm polyester/12 μm aluminium foil/75 μm cast polypropylene/15.0 μm biaxially-oriented nylon) with a dimension of 14 cm × 18 cm × 4 cm were used to fill the product (Fig 1C). Each pouch was filled manually with 200 grams of the product, and then air from the head space was squeezed out so that the pouches can be sealed using a Henkovac type H 1502 packaging machine ('s-Hertogenbosch, the Netherlands). The filled and sealed pouches were kept on perforated trays and subsequently transferred into a retort vessel. A pilot-scale horizontal stationary retort system (Owner Foods Machinery Co., Ltd., Bangkok, Thailand) was used (Fig 1D). For thermal processing, the retort temperature was kept constant at 121˚C and the processing time was 54 min with an $F_0$ value of 3.57 min calculated using Ball's method [20]. Pressure was maintained at 1.83 kg/cm$^2$ throughout the process using a hot water-air mixture. Temperatures in the retort and pouch center were continuously monitored with thermocouples (Fig 1E), and the data was recorded using a data logging device (EVal Flex System, ELLAB, Denmark). The temperature input data was converted into corresponding process lethality values, which were expressed as $F_0$ values [20].

After processing the pouches to the required $F_0$ value, the pouches were cooled rapidly by recirculating water to 45˚C. The processed retort pouches were stored in a dry place at room temperature (28-30˚C), and the samples were subjected to analyses including chemical (moisture, $a_w$, pH, and peroxide value (PV)), physical (creaming layer, viscosity, and color), and sensory (color, viscosity, odor and overall liking) properties. An optimum stabilizer was selected for further study in combination with antioxidants.

## Effects of antioxidants on retorted southern-style *Pad Thai* sauce

In this study, retorted southern-style *Pad Thai* sauce was prepared in the presence of 2.3% PS + 0.1% XG as previously described. Different antioxidants including 500 mg/kg ascorbyl palmitate plus 500 mg/kg α-tocopherol (As + Toc), 100 mg/kg EDTA, 200 mg/kg BHT and 500 mg/kg ascorbyl palmitate + 500 mg/kg α-tocopherol + 100 mg/kg EDTA + 200 mg/kg BHT (mixed antioxidants) were incorporated in order to determine their effects on the oxidative and color stabilities of the product. The final concentration of all antioxidants in the mixed antioxidants was the same for each one used separately. The resulting sauces were subjected to analyses for PV and color. The appearance of the resulting sauces is shown in Fig 1F.

## Chemical analysis

**Determination of moisture content, $a_w$ and pH.**   The AOAC method number 950.46 was used for moisture content analysis in the retorted southern-style *Pad Thai* sauce [21]. The $a_w$ was determined at 25˚C using an Aqualab Series 3TE $a_w$ meter (Decagon, Pullman, WA, USA). The pH was measured using a calibrated pH meter (Cyberscan 500, Singapore).

**Determination of PV.**   PV was determined according to the method of Low and Ng [22]. The sample (1 g) was treated with 25 mL of a chloroform:acetic acid mixture at 2:3 ratio. The mixture was shaken vigorously, followed by the addition of 1 mL of saturated potassium

iodide. The mixture was kept in the dark for 5 min, and 75 mL of distilled water was added. A 0.5 mL starch solution (1%, w/v) was added as an indicator. The PV was determined by titrating the iodine liberated from potassium iodide with a standardized 0.01 N sodium thiosulfate solution. The PV was expressed as milliequivalents of free iodine/kg sample.

## Physical analysis

**Creaming layer.** A creaming layer was determined using a modified method of Agboola et al. [5], where 15 mL of freshly retorted southern-style *Pad Thai* sauce was poured into a glass cylinder. After 24 h of storage at 25°C, the separated cream layer was read and the creaming layer was calculated according to Eq (1).

$$Creaming\ layer\ (\%) = (Volume\ of\ cream\ layer/Volume\ of\ total\ emulsion) \times 100 \qquad (1)$$

**Viscosity measurement.** Viscosity of the retorted southern-style *Pad Thai* sauce was determined at 25°C using a Brookfield Viscometer Model RV-DV II Pro$^+$ (Brookfield Engineering Inc, Middleborough, MA, USA) at 200 rpm. A UL cylindrical spindle (ULA-15E) was used along with a Brookfield Ametek UL adapter.

**Determination of color.** Colorimetric values of the retorted southern-style *Pad Thai* sauce were obtained in triplicate using a portable Hunterlab Miniscan/EX instrument (10° standard observers, illuminant D65, Hunter Assoc. Laboratory, VA, USA). The instrument was calibrated to a white and black standard. The tristimulus $L^*$ (lightness), $a^*$ (redness/greenness), and $b^*$ (yellowness/blueness) measurement mode was used as it relates to the human eye response to color. Total difference in color ($\Delta E^*$) was then calculated according to Eq (2) [23]

$$\Delta E* = \sqrt{\Delta L*^2 + \Delta a*^2 + \Delta b*^2} \qquad (2)$$

where $\Delta L^*$, $\Delta a^*$, and $\Delta b^*$ are the differences between the corresponding color parameter of the sample and that of the original control sauce without retorting ($L^* = 31.98$, $a^* = 19.41$, and $b^* = 38.22$).

## Sensory analysis

Sensory properties of the retorted southern-style *Pad Thai* sauce including color, viscosity, odor and, overall liking were evaluated by 30 untrained panelists using the 9-point hedonic scale (a score of 1 = not like very much, 5 = neither like nor dislike, and 9 = like extremely) [24]. The samples were labelled with random three-digit codes. The panelists were instructed to rinse their mouth with water after each sample evaluation, and the order of presentation of the samples was randomized [25].

## Statistical analysis

Three different lots for each type of samples were produced (n = 3). Data was presented as means ± standard deviation from triplicate determinations. Statistical comparisons were performed using the One-Way Analysis of Variance (ANOVA) statistics method and by Duncan's new multiple range test (DMRT) by SPSS software. *P*-value < 0.05 was considered as significantly different [26].

## Results and discussion

### Chemical property of retorted southern-style *Pad Thai* sauce added with different stabilizers

Chemical property of the retorted southern-style *Pad Thai* sauce added with different stabilizers is shown in Table 1. The control (without added stabilizer) had the highest moisture

**Table 1. Chemical property of retorted southern-style *Pad Thai* sauce added with different stabilizers.**

| Sample | Moisture (%) | $a_w$ | pH | PV (meq/kg sample) |
|---|---|---|---|---|
| Control | 52.28±1.35[c] | 0.97±0.00[b] | 4.97±0.01[c] | 19.60±0.05[b] |
| PS + XG | 45.97±0.28[b] | 0.95±0.00[a] | 4.95±0.00[b] | 23.03±5.78[b] |
| LT | 45.03±1.77[b] | 0.95±0.00[a] | 4.38±0.01[a] | 9.77±0.04[a] |
| WPI | 40.17±0.15[a] | 0.95±0.00[a] | 5.01±0.00[d] | 19.65±0.33[b] |

PS + XG = 2.3% potato starch + 0.1% xanthan gum, LT = 0.5% lecithin and WPI = 4% whey protein isolate.

Values are given as mean ± standard deviation from triplicate determinations.

Different letters in the same column indicate significant differences (p<0.05).

content and $a_w$ (p<0.05). The addition of stabilizers led to a reduction of moisture content and $a_w$. The lowest moisture content was found in the sauce stabilized by WPI, followed by LT and PS mixed with XG. Protein-based stabilizers may undergo gelation upon retorting and the moisture could be entrapped in the gel network. Carbohydrate hydrocolloids (PS and XG) could form a composite gel and hold water as well, whereas lipid-based stabilizers (e.g. lecithin) could bind water with their polar part. Both chemical binding and physical entrapping resulted in the increased water holding capacity of the sauce, which was difficult to evaporate upon the determination of moisture and $a_w$. The pH values of the sauces were altered by the stabilizers. When compared to the control, PS mixed with XG led to a negligible decrease in pH, whereas LT caused a gradual decrease. However, WPI showed a different trend in which it caused an increase in pH of the sauce (p<0.05). The highest pH of the sauce stabilized by WPI would result in the highest water holding capacity of the sauce. The PV of the control sauce was similar to the stabilized sauces with PS mixed with XG and WPI (p>0.05), whereas the sauce stabilized by LT had about 50% reduction in PV. This was probably due to the antioxidant activity of LT. It has been reported that LT can function as an antioxidant in emulsion system [14], vegetable oil [27], and margarine [28]. Pan et al. [14] reported that the antioxidant activity of LT reduced permeation of free radicals across the emulsion interface. XG has been reported to show some antioxidant activity in emulsion [29], but it could not be seen in this study. This was probably due to the different emulsion systems and different concentrations applied.

## Physical property of retorted southern-style *Pad Thai* sauce added with different stabilizers

Stability, viscosity, and color generally contribute to the acceptance of a sauce [6]. From the results, creaming layer can be observed in the control sauce after retorting (Table 2), whereas no creaming layer was detected in the sauces stabilized by all the stabilizers used (Table 2). With the stabilizers, emulsion should be more stabilized especially by increasing the apparent viscosity of the system (Table 2). The apparent viscosity of the sauce markedly increased when the stabilizers were used. Mixed carbohydrate hydrocolloids (PS + XG) rendered the sauce with the highest viscosity, followed by WPI and LT. Polysaccharides are mainly used as thickening agents for modifying the texture of the food system and enhancing the viscosity of the aqueous phase, which increase emulsion stability, particularly towards creaming [7]. Polysaccharides play important roles as thickening, stabilizing, and gelling agents in many foods. Also, for emulsion systems, polysaccharides are very often used to improve emulsion stability and textural properties [30]. However, the influence of polysaccharides on the stability of emulsions depends on the concentration of polysaccharides and the characteristics of the polysaccharides and emulsion system [9]. XG has relatively high water binding capacity and high

**Table 2. Physical property of retorted southern-style *Pad Thai* sauce added with different stabilizers.**

| Sample | Creaming layer (%) | Apparent viscosity (mPa.s) | Color | | | |
|---|---|---|---|---|---|---|
| | | | $L^*$ | $a^*$ | $b^*$ | $\Delta E^*$ |
| Control | 10±1 | 25±1[a] | 29.52±0.35[a] | 22.22±0.20[a] | 38.32±0.85[a] | 3.88±0.82[a] |
| PS + XG | ND[#] | 14167±205[d] | 31.48±0.07[c] | 23.14±0.21[b] | 41.52±0.93[b] | 5.09±0.80[b] |
| LT | ND | 2738±18[b] | 30.60±0.12[b] | 23.37±0.35[b] | 39.60±0.48[a] | 4.48±0.14[ab] |
| WPI | ND | 5697±35[c] | 29.33±0.20[a] | 22.23±0.28[a] | 38.21±0.99[a] | 3.91±0.11[a] |

PS + XG = 2.3% potato starch + 0.1% xanthan gum, LT = 0.5% lecithin and WPI = 4% whey protein isolate.

Values are given as mean ± standard deviation from triplicate determinations.

Different letters in the same column indicate significant differences (p<0.05).

$\Delta E^*$ was calculated relative to the color of the original control sauce without retorting. $L^*$, $a^*$, and $b^*$ of the original control sauce without retorting were 31.98±0.06, 19.41±0.20, and 38.22±0.54, respectively.

[#]ND, not detected.

viscosity of aqueous solutions [31]. Krystyjan et al. [7] reported the positive effect of combining XG with PS on the texture and acceptability of caramel sauce. Whey proteins can play a dual role in heat-treated emulsion by acting as an emulsifier and stabilizer [8] due to their hydrophobic and hydrophilic segments within the polypeptide chain. LTs are important ingredients in the commercial manufacturing of emulsions. Commercial LT has increased the heat stability of recombined milk systems [13]. The addition of 2.5% (w/w) LT slightly improved the stability of the retorted emulsions [5]. Many studies have been carried out on LTs reporting on their surface active properties, their competition with proteins at oil/water interfaces and their interactions with proteins [5].

For the color of the retorted sauces, changes in $L^*$ and $a^*$ values were found when stabilizers were added (p<0.05). The sauces stabilized by PS + XG and by LT were lighter and redder than the control and the WPI-stabilized sauce. The PS + XG-stabilized sauce had a higher $b^*$ value than other sauces (p<0.05). The total difference in color ($\Delta E^*$) was calculated relative to the color of the original control sauce without retorting. Only the PS + XG-stabilized sauce had a higher $\Delta E^*$ than the control. This was due to an increase in $L^*$, $a^*$, and $b^*$ of that sauce after retorting. However, the $\Delta E^*$ of all sauces was quite low, suggesting a similar color of the sauces with and without stabilizers, which was related to the sensory analysis (Table 3). Carbohydrate hydrocolloids may undergo thermal degradation and the monomers released can further react with amine groups in the sauce to form brown pigments via Maillard reaction as indicated by an increase in the $a^*$ and $b^*$ values. XG was found to be less thermally stable than starch [32]. XG is a heteropolysaccharide made up of the building blocks of D-glucose, D-mannose, and D-glucuronic acid residues [12].

**Table 3. Sensory property of retorted southern-style *Pad Thai* sauce added with different stabilizers.**

| Sample | Color | Viscosity | Odor | Overall liking |
|---|---|---|---|---|
| Control | 6.87±1.22[a] | 4.09±0.64[a] | 6.47±1.66[a] | 6.47±1.66[a] |
| PS + XG | 7.03±0.98[a] | 6.60±1.07[b] | 6.77±0.97[a] | 6.80±1.00[a] |
| LT | 7.00±0.83[a] | 6.83±1.18[b] | 7.00±1.12[a] | 7.13±1.11[a] |
| WPI | 7.15±1.03[a] | 6.70±1.06[b] | 6.97±1.19[a] | 7.25±1.08[a] |

PS + XG = 2.3% potato starch + 0.1% xanthan gum, LT = 0.5% lecithin and WPI = 4% whey protein isolate.

Values are given as mean ± standard deviation from 30 determinations.

Different letters in the same column indicate significant differences (p<0.05).

## Sensory property of retorted southern-style *Pad Thai* sauce added with different stabilizers

Sensory property of the sterilized southern-style *Pad Thai* sauce added with different stabilizers is shown in Table 3. Panelists rated all sauces with the same range of scores for color, odor, and overall liking (p<0.05). All samples showed acceptability with the same score for likeness being equal to 7 (moderately like) for all attributes tested except for the viscosity of the control (score ~4). The viscosity scores of the stabilizer-added sauces were higher than the control. The instrumental color values were different among the samples, but the panelists scored the color similarly. Overall, the stabilizers used did not affect the odor and overall liking of the resulting sauces. Thus, the sauce stabilized by PS + XG was selected for further study due to its highest apparent viscosity.

## Effects of antioxidants on lipid oxidation and color of retorted southern-style *Pad Thai* sauce

Effects of the antioxidants on the PV of the retorted southern-style *Pad Thai* sauce is shown in Fig 2. The control sauce had the highest PV, followed by the sauces with As + Tc, EDTA, BHT, and mixed antioxidants (p<0.05). Generally, lipid oxidation in the emulsion is triggered by permeation of free radicals generated at the emulsion interface [10, 11, 14]. Thus, the polarity of free radicals may influence their movement to the oil or water phase. As + Tc and BHT are lipid-solubilizing antioxidants that may function as free radical scavengers mainly in the oil phase, but their activities may be different to some degree. Ascorbyl palmitate is a potent antioxidant in protecting lipids from peroxidation and is a free radical scavenger [33]. BHT is fairly heat stable and is used in heat-processed foods [34]. Thus, BHT may show a better carry-through property than As + Tc in the retorted southern-style *Pad Thai* sauce, which can

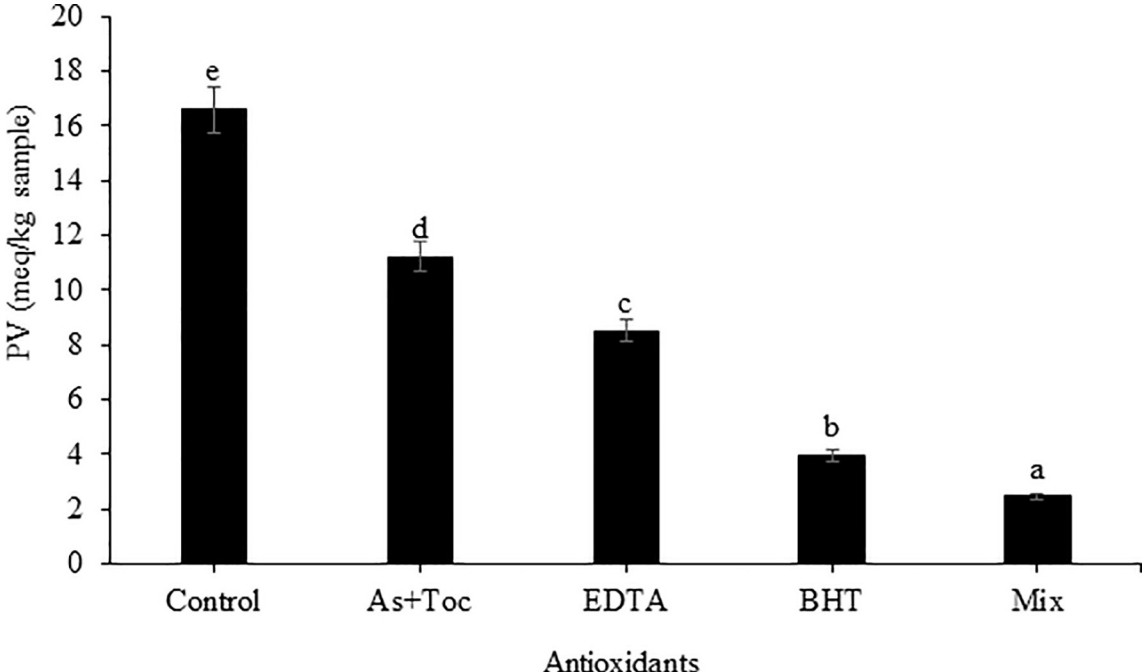

**Fig 2. PV of retorted southern-style *Pad Thai* sauce added with different antioxidants.** Different letters on the bars indicate significant differences (p<0.05). As + Toc = 500 mg/kg ascorbyl palmitate + 500 mg/kg α-tocopherol, EDTA = 100 mg/kg EDTA, BHT = 200 mg/kg BHT, and Mix = 500 mg/kg ascorbyl palmitate + 500 mg/kg α-tocopherol + 100 mg/kg EDTA + 200 mg/kg BHT.

**Table 4. Color of retorted southern-style *Pad Thai* sauce added with different antioxidants.**

| Sample | $L^*$ | $a^*$ | $b^*$ | $\Delta E^*$ |
|---|---|---|---|---|
| Control | 29.95 ± 0.15[d] | 24.66 ± 0.18[b] | 43.37 ± 0.66[c] | 7.28±0.22[a] |
| As + Toc | 28.45 ± 0.31[b] | 29.28 ± 0.33[d] | 44.13 ± 0.94[c] | 11.20±0.24[c] |
| EDTA | 30.17 ± 0.08[e] | 30.03 ± 0.95[e] | 43.92 ± 0.60[c] | 11.29±0.37[c] |
| BHT | 29.31 ± 0.27[c] | 27.82 ± 0.32[c] | 42.02 ± 1.68[b] | 8.63±0.62[b] |
| Mix | 23.67 ± 0.03[a] | 23.67 ± 0.03[a] | 35.57 ± 0.13[a] | 8.51±0.19[b] |

As + Toc = 500 mg/kg ascorbyl palmitate + 500 mg/kg α-tocopherol, EDTA = 100 mg/kg EDTA, BHT = 200 mg/kg BHT, and Mix = 500 mg/kg ascorbyl palmitate + 500 mg/kg α-tocopherol + 100 mg/kg EDTA 200 mg/kg BHT.

$\Delta E^*$ was calculated relative to the color of the original control sauce without retorting. $L^*$, $a^*$, and $b^*$ of the original control sauce without retorting were 31.67±0.04, 21.01±0.14, and 37.31±0.76, respectively.

Values are given as mean ± standard deviation from triplicate determinations.

Different letters in the same column indicate significant differences ($p<0.05$).

pronounce superior antioxidant activity as indicated by a lower PV compared to As + Tc. EDTA may chelate the metal ion presented in the water phase due to its water solubility and chelating ability. Mixed antioxidants showed the best activity in retarding the PV, which was due to the presence of antioxidants in both the oil and aqueous phases with different modes of action. Synergistic mechanism among antioxidants may lead to the lowest PV.

Effects of antioxidants on the color of the sterilized southern-style *Pad Thai* sauce is shown in Table 4. Generally, the $L^*$, $a^*$, and $b^*$ values changed considerably when the antioxidants were added. The $L^*$ value of the sauce with EDTA was higher than the control, whereas other treatments were lower ($p<0.05$). The lowest $L^*$ value was noticeable in the sauce with the mixed antioxidants ($p<0.05$). For the $a^*$ value, the sauces with the antioxidants except for the mixed antioxidants had a higher $a^*$ value than the control ($p<0.05$). The sauce with the mixed antioxidants showed the most similar $a^*$ value to the control. For the $b^*$ value, the control sauce had a similar $b^*$ value to the sauces with As + Toc and EDTA ($p>0.05$), whereas the $b^*$ values slightly decreased in the presence of BHT ($p<0.05$). The $b^*$ value of the sauce markedly decreased when the mixed antioxidants were used ($p<0.05$). For the color difference ($\Delta E^*$) which was calculated against the original control sauce without retorting, it was found that the control sauce had the lowest $\Delta E^*$ value. Among the samples with the antioxidants, the mixed antioxidants and BHT-added sauces had lower $\Delta E^*$ values than the As + Toc and EDTA-treated ones. Thus, based on antioxidant activity and color stability, the mixed antioxidants were selected as the optimum antioxidant for the sterilized southern-style *Pad Thai* sauce.

## Conclusion

Retorted southern-style *Pad Thai* sauce was successfully produced, and it can be stabilized by 2.3% PS + 0.1% XG in combination with mixed antioxidants during retorting. This sauce stabilized by 2.3% PS + 0.1% XG showed a desirable viscosity, water holding capacity, and color without any creaming layer or detrimental effects on sensory properties. The mixed antioxidants showed the best activity in retarding the PV and $\Delta E^*$ values, which was due to the presence of antioxidants in both the oil and aqueous phases. Thus, the use of an appropriate stabilizer and antioxidant can be a strategy in improving textural property and preventing lipid oxidation of retorted southern-style *Pad Thai* sauce. However, storage test should be done in the future to ensure the stability of the product.

## Supporting information

**S1 Table. Chemical property of retorted southern-style *pad thai* sauce added with different stabilizers.**
(DOCX)

**S2 Table. Physical property of retorted southern-style *pad thai* sauce added with different stabilizers.**
(DOCX)

**S3 Table. Sensory property of retorted southern-style *pad thai* sauce added with different stabilizers.**
(DOCX)

**S4 Table. Color of retorted southern-style *pad thai* sauce added with different antioxidants.**
(DOCX)

**S1 Fig.**
(PPTX)

**S2 Fig.**
(PPTX)

## Acknowledgments

This research was partially supported by the New Strategic Research (P2P) project, Walailak University, Thailand.

## Author Contributions

**Conceptualization:** Worawan Panpipat, Manat Chaijan.

**Data curation:** Yanyong Cheenkaew, Worawan Panpipat, Manat Chaijan.

**Funding acquisition:** Manat Chaijan.

**Investigation:** Yanyong Cheenkaew.

**Methodology:** Worawan Panpipat, Manat Chaijan.

**Supervision:** Worawan Panpipat, Manat Chaijan.

**Writing – original draft:** Yanyong Cheenkaew, Worawan Panpipat, Manat Chaijan.

**Writing – review & editing:** Worawan Panpipat, Manat Chaijan.

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
