## [Decision Letter · Decision Letter 0]

14 Apr 2020

PONE-D-19-34660

Southern style Pad Thai sauce: from traditional culinary treat to convenience food in retortable pouches

PLOS ONE

Dear Dr. Chaijan,

Thank you for submitting your manuscript to PLOS ONE. After careful consideration, we feel that it has merit but does not fully meet PLOS ONE’s publication criteria as it currently stands. Therefore, we invite you to submit a revised version of the manuscript that addresses the points raised during the review process.

We would appreciate receiving your revised manuscript by May 29 2020 11:59PM. To enhance the reproducibility of your results, we recommend that if applicable you deposit your laboratory protocols in protocols.io, where a protocol can be assigned its own identifier (DOI) such that it can be cited independently in the future. For instructions see: http://journals.plos.org/plosone/s/submission-guidelines#loc-laboratory-protocols

We look forward to receiving your revised manuscript.

Kind regards,

Edy de Brito

Academic Editor

PLOS ONE

Additional Editor Comments (if provided):

Dear author,

Finally, we were able to have the required reviewers comments. Once you supply appropriate point by point answer to their queries I will be glad to revaluate your manuscript.

2. In your Methods, please state the exact source of all ingredients used in the preparation of the Pad Thai sauce.

Reviewers' comments:

Reviewer's Responses to Questions

**Comments to the Author**

1. Is the manuscript technically sound, and do the data support the conclusions?

Reviewer #1: Partly

Reviewer #2: Yes

2. Has the statistical analysis been performed appropriately and rigorously? 

Reviewer #1: Yes

Reviewer #2: Yes

3. Have the authors made all data underlying the findings in their manuscript fully available?

Reviewer #1: Yes

Reviewer #2: Yes

4. Is the manuscript presented in an intelligible fashion and written in standard English?

Reviewer #1: No

Reviewer #2: Yes

5. Review Comments to the Author

Reviewer #1: The study is very interesting because it relates culture (gastronomy) to food science and technology. It focuses on the availability of an important sauce in Thailand to other cities and countries with good sensory quality. Some improvements are necessary to better comprehension of data and conclusions. English revision is necessary and I will point out some mistakes, but other need to be fixed.

Abstract:

The objective is missing in the abstract.Also, it is missing information related to methodology. which stabilizers were used? and antioxidants?

For key words, choose or stabilizer or stabilization

Introduction

Line 2 - take out the comma before differs - subject and verb come together

Lines 66 and 67 - Based on what? Is there a study with Pad Thai sauce? Is there a study relating the sauce and lipid oxidation? little is shown in this part for oxidation of sauces.

take out this sentence: Xanthan gum was approved by the FDA for use as a stabilizer, emulsifier, thickener, suspending agent, bodying agent, or foam enhancer in foods [10].

This information is not necessary in the introduction.

Make objective simpler, take out from in order to.

References must be improved with newer articles related to the study

Methods:

the way procedures are described, it is very confusing.

As a suggestion starting writing everything related to the production of the sauce including the retort procedures.

Then present the additions of stabilizers and antioxidants - differences, percentages. all the possibilities. Follow with all the chemical analysis and close with sensory and statistical analysis.

For the sauce preparation: were the coconut milk and chili and shrimp paste industrialized? or were they prepared by researchers? how?

line 101 - take out respectively

It is not clear how stabilizers were added, separately, together, how many samples. only at the results section, we understand the number of samples.

The mix with all the antioxidants was prepared with the same amount of each one used separately?

For the sensory analysis, why 30 panelists? is this number enough? refer to a study.

How many samples each panelist tasted in one session?

For statistical analysis - authors state 3 different lots of retorted southern style Pad Thai sauce. three lots for each type of sample?

Results and discussion

Lines 192, 193 - PV was similar, how with p<0.05?

no discussion fo pH and Aw. include

197 - 198 - rewrite sentence. it is not understandable

It is strange that pH so similar are all statistically different. Can you check this information.

line 221 - no creamy layer was not were

Line 241 - Table shows statistical differences so it is not negligible changes as written.

line 247 - Which differences may not be detected by consumers? Relate with sensory analysis.

line 255 - To investigate the effect of different stabilizers on sensory properties, a nine-point hedonic scale

256 was used to evaluate the likeness of retorted southern style Pad Thai sauce. take out this part. it is methodology

line 262 - rewrite - English problems

Line 265 - Why did you select PS + XG? based on sensory, it could be any of them.

What are the limitations of the study? include

Reviewer #2: The present work on "Southern style Pad Thai sauce: from traditional culinary treat to convenience food in retortable pouches" is interesting. The work has been executed meticulously and the results have been expressed aptly. However, for the benefit of the international readers, it is recommended to share pictures of "Pad Thai sauce" containing without and with stabilizers and antioxidants. Further, since this product is meant for long term storage, results of the effect of the optimized ingredients on the storage stability should have incorporated at least under accelerated storage conditions.

6. PLOS authors have the option to publish the peer review history of their article (what does this mean?). If published, this will include your full peer review and any attached files.

Reviewer #1: No

Reviewer #2: Yes: Dr. Narender Raju Panjagari

---

## [Author Response · Author response to Decision Letter 0]

22 Apr 2020

Response to Reviewers

All points raised by the academic editor and reviewers were carefully addressed and answered point-by-point. A revision was made in highlighted red fonts. The revised manuscript was carefully proofread for English by a native speaker, Mr. Dave Chang from USA. The revised manuscript was carefully prepared to meet PLOS ONE's style requirements.

Additional Editor Comments (if provided):

Dear author,

Finally, we were able to have the required reviewers comments. Once you supply appropriate point by point answer to their queries I will be glad to revaluate your manuscript.

Ans: The revised manuscript was carefully prepared to meet PLOS ONE's style requirements.

2. In your Methods, please state the exact source of all ingredients used in the preparation of the Pad Thai sauce.

 Ans: The exact sources of all ingredients used in the preparation of the Pad Thai sauce were added “All ingredients used in the preparation of the sauce were obtained from a local market in Thasala, Nakhon Si Thammarat.”

Reviewer #1: The study is very interesting because it relates culture (gastronomy) to food science and technology. It focuses on the availability of an important sauce in Thailand to other cities and countries with good sensory quality. Some improvements are necessary to better comprehension of data and conclusions. English revision is necessary and I will point out some mistakes, but other need to be fixed.

Ans: The revised manuscript was carefully proofread for English by a native speaker, Mr. Dave Chang from USA.

Abstract:

The objective is missing in the abstract. Also, it is missing information related to methodology. which stabilizers were used? and antioxidants?

For key words, choose or stabilizer or stabilization

Ans: The objective was added in the abstract “Thus, the objective of this study was to investigate the effects of stabilizers and antioxidants on the stability of retorted southern-style Pad Thai sauce.” The stabilizers and antioxidants used were also mentioned in the abstract “Southern-style Pad Thai sauce was prepared according to the traditional recipe in the presence of different stabilizers (2.3% potato starch (PS) + 0.1% xanthan gum (XG), 0.5% soy lecithin (LT), and 4% whey protein isolate (WPI)) and antioxidants (500 mg/kg ascorbyl palmitate + 500 mg/kg �-tocopherol (As + Toc), 100 mg/kg ethylenediaminetetraacetic acid (EDTA), 200 mg/kg butylated hydroxytoluene (BHT), and As+Toc+EDTA+BHT (mixed antioxidants))..” For the keywords, “stabilizer” was selected and “stabilization” was deleted.

Introduction

Line 2 - take out the comma before differs - subject and verb come together

Ans: Done as recommended. 

Lines 66 and 67 - Based on what? Is there a study with Pad Thai sauce? Is there a study relating the sauce and lipid oxidation? little is shown in this part for oxidation of sauces.

Ans: The assumption was based on the previous reports in other emulsified sauces. We also observed during the preliminary test after retorting the southern style Pad Thai sauce. Here is the detail we provided in the revision “However, based on the emulsion-based system that produces other emulsified sauces [5, 6, 7, 8, 9], the retort process of southern-style Pad Thai sauce may encounter phase separation and lipid oxidation during retorting. It was reported that the retorting of emulsion at high temperatures (121�C for 16 min) resulted in lipid oxidation and an increase in coalescence of oil droplets [5]. Lipid oxidation is a critical issue in regard to emulsified foods because it is considered as an interfacial phenomenon [10, 11]. The large interfacial area of emulsion provides numerous sites for lipid oxidation to occur [10, 11]. Also, the separation of the solid and liquid phase is one of the main problems associated with the quality of sauces [6].”

take out this sentence: Xanthan gum was approved by the FDA for use as a stabilizer, emulsifier, thickener, suspending agent, bodying agent, or foam enhancer in foods [10].

This information is not necessary in the introduction.

Ans: Done as recommended.

Make objective simpler, take out from in order to.

Ans: Done as recommended.

References must be improved with newer articles related to the study

Ans: It is difficult to find the new articles with related references because there is no report on the production of characteristics of southern style Pad Thai sauce, especially the heat-treated one. So, we tried our best to find the related references from other sauces which published from 1992-2018. We have added two more updated references in 2019 and 2020 regarding the lipid oxidation of emulsion in general. “Lipid oxidation is a critical issue in regard to emulsified foods because it is considered as an interfacial phenomenon [10, 11]. The large interfacial area of emulsion provides numerous sites for lipid oxidation to occur [10, 11].” 

10. Mirzanajafi-Zanjani M, Yousefi M, Ehsani A. Challenges and approaches for production of a healthy and functional mayonnaise sauce. Food Sci Nutr.2019;7:2471-2484.

11. Noon J, Mills TB, Norton IT. The use of natural antioxidants to combat lipid oxidation in O/W emulsions. J Food Eng.2020:110006.

Methods:

the way procedures are described, it is very confusing.

As a suggestion starting writing everything related to the production of the sauce including the retort procedures.

Then present the additions of stabilizers and antioxidants - differences, percentages. all the possibilities. Follow with all the chemical analysis and close with sensory and statistical analysis.

Ans: The procedures stared with the production of southern-style Pad Thai sauce, the effect of stabilizer on retorted southern-style Pad Thai sauce including the retort procedures, the effect of antioxidants on retorted southern-style Pad Thai sauce, the chemical analysis (moisture content, aw, pH, and PV), the physical analysis (creaming layer, viscosity, and color), sensory analysis, and statistical analysis.

For the sauce preparation: were the coconut milk and chili and shrimp paste industrialized? or were they prepared by researchers? how?

Ans: We added in this section that “All ingredients used in the preparation of the sauce were obtained from a local market in Thasala, Nakhon Si Thammarat.” as recommended by the editor. In Thailand, coconut milk, chili paste and shrimp paste can be purchased in the market.

line 101 - take out respectively

Ans: Done as recommended.

It is not clear how stabilizers were added, separately, together, how many samples. only at the results section, we understand the number of samples.

Ans: We added the detail of samples in the section of Effects of stabilizers on retorted southern-style Pad Thai sauce “To study the effects of stabilizers on retorted southern-style Pad Thai sauce, different stabilizers including 2.3% PS plus 0.1% XG, 0.5% LT, and 4% WPI were separately added at the stir-frying step during the preparation of the sauce to obtain four different sauces – namely, the control (without stabilizer), sauce with PS + XG, sauce with LT, and sauce with WPI.”

The mix with all the antioxidants was prepared with the same amount of each one used separately?

Ans: Yes, the final concentration of all antioxidants in the mixed antioxidants was the same of each one used separately. We also stated this issue in the Material and methods that “Different antioxidants including 500 mg/kg ascorbyl palmitate plus 500 mg/kg �-tocopherol (As + Toc), 100 mg/kg EDTA, 200 mg/kg BHT and 500 mg/kg ascorbyl palmitate + 500 mg/kg �-tocopherol + 100 mg/kg EDTA + 200 mg/kg BHT (mixed antioxidants) were incorporated in order to determine their effects on the oxidative and color stabilities of the product. The final concentration of all antioxidants in the mixed antioxidants was the same for each one used separately.”

For the sensory analysis, why 30 panelists? is this number enough? refer to a study.

Ans: In this study we used 30 panelists. From the textbook reference (Watts, B. M., Ylimaki, G. L., Jeffery, L. E., & Elias, L. G. (1989). Basic sensory methods for food evaluation. IDRC, Ottawa, ON, CA.), the in-house consumer panels (pilot consumer panels) usually consist of 30 to 50 untrained panelists selected from personnel within the organization where the product development or research is being conducted. We also added this reference in the revised version.

How many samples each panelist tasted in one session?

Ans: There were 4 samples (control, PS + XG, LT, and WPI) each panelist tasted in one session.

For statistical analysis - authors state 3 different lots of retorted southern style Pad Thai sauce. three lots for each type of sample?

Ans: Yes, it is three lots for each type of sample. We also stated in the statistical analysis “Three different lots for each type of samples were produced (n = 3)."

Results and discussion

Lines 192, 193 - PV was similar, how with p<0.05?

Ans: It was changed to “p>0.05”.

no discussion fo pH and Aw. Include

Ans: The pH was originally discussed “The pH values of the sauces were altered by the stabilizers. When compared to the control, PS mixed with XG led to a negligible decrease in pH, whereas LT caused a gradual decrease. However, WPI showed a different trend in which it caused an increase in pH of the sauce (p<0.05). The highest pH of the sauce stabilized by WPI would result in the highest water holding capacity of the sauce.” For the aw, the discussion was extended. “The control (without added stabilizer) had the highest moisture content and aw (p<0.05). The addition of stabilizers led to a reduction of moisture content and aw. The lowest moisture content was found in the sauce stabilized by WPI, followed by LT and PS mixed with XG. Protein-based stabilizers may undergo gelation upon retorting and the moisture could be entrapped in the gel network. Carbohydrate hydrocolloids (PS and XG) could form a composite gel and hold water as well, whereas lipid-based stabilizers (e.g. lecithin) could bind water with their polar part. Both chemical binding and physical entrapping resulted in the increased water holding capacity of the sauce, which was difficult to evaporate upon the determination of moisture and aw.”

197 - 198 - rewrite sentence. it is not understandable

Ans: It was rewritten “Pan et al. [14] reported that the antioxidant activity of LT reduced permeation of free radicals across the emulsion interface. XG has been reported to show some antioxidant activity in emulsion [29], but it could not be seen in this study.”

It is strange that pH so similar are all statistically different. Can you check this information.

Ans: We rechecked the information regarding the pH and they were all statistically different. The means of the pH were quite similar but the standard deviations were very low. Therefore, they were statistically different. 

line 221 - no creamy layer was not were

Ans: Done as recommended.

Line 241 - Table shows statistical differences so it is not negligible changes as written.

Ans: We deleted the term “negligible” and the p value was added. “For the color of the retorted sauces, changes in L* and a* values were found when stabilizers were added (p<0.05).”

line 247 - Which differences may not be detected by consumers? Relate with sensory analysis.

Ans: The sentence was changed to “However, the �E* of all sauces was quite low, suggesting a similar color of the sauces with and without stabilizers, which was related to the sensory analysis (Table 3).”

line 255 - To investigate the effect of different stabilizers on sensory properties, a nine-point hedonic scale was used to evaluate the likeness of retorted southern style Pad Thai sauce. take out this part. it is methodology

Ans: Done as recommended. 

line 262 - rewrite - English problems

Ans: The sentence and the issue about the viscosity score were rewritten. 

Line 265 - Why did you select PS + XG? based on sensory, it could be any of them.

What are the limitations of the study? Include

Ans: In this study we selected PS+XG because it can improve the apparent viscosity to the highest extent without a negative effect on sensory characteristic. “Thus, the sauce stabilized by PS + XG was selected for further study due to its highest apparent viscosity.”

Reviewer #2: The present work on "Southern style Pad Thai sauce: from traditional culinary treat to convenience food in retortable pouches" is interesting. The work has been executed meticulously and the results have been expressed aptly. However, for the benefit of the international readers, it is recommended to share pictures of "Pad Thai sauce" containing without and with stabilizers and antioxidants. Further, since this product is meant for long term storage, results of the effect of the optimized ingredients on the storage stability should have incorporated at least under accelerated storage conditions.

Ans: The pictures relating to the Pad Thai sauce are shown in Fig. 1 a-f. 

“Fig. 1 Southern-style Pad Thai, so-called Pak Phanang Mee Pad (spicy stir-fried rice noodles with coconut milk) (a), preparation of southern-style Pad Thai sauce (b), retortable pouch used to fill the product (c), pilot-scale horizontal stationary retort system (d), monitoring the temperatures in the retort and pouch center with thermocouples (e), and appearance of the final retorted sauce (f).”

For the storage test, unfortunately, it was not done yet in the present study. However, in the conclusion, we recommended to do the storage test for the further research as recommended. “However, storage test should be done in the future to ensure the stability of the product.”

---

## [Decision Letter · Decision Letter 1]

5 May 2020

Southern-style Pad Thai sauce: from traditional culinary treat to convenience food in retortable pouches

PONE-D-19-34660R1

Dear Dr. Chaijan,

We are pleased to inform you that your manuscript has been judged scientifically suitable for publication and will be formally accepted for publication once it complies with all outstanding technical requirements.

With kind regards,

Edy de Brito

Academic Editor

PLOS ONE

Additional Editor Comments (optional):

Reviewers' comments:

Reviewer's Responses to Questions

**Comments to the Author**

1. If the authors have adequately addressed your comments raised in a previous round of review and you feel that this manuscript is now acceptable for publication, you may indicate that here to bypass the “Comments to the Author” section, enter your conflict of interest statement in the “Confidential to Editor” section, and submit your "Accept" recommendation.

Reviewer #1: All comments have been addressed

Reviewer #2: All comments have been addressed

2. Is the manuscript technically sound, and do the data support the conclusions?

Reviewer #1: Yes

Reviewer #2: Yes

3. Has the statistical analysis been performed appropriately and rigorously? 

Reviewer #1: Yes

Reviewer #2: Yes

4. Have the authors made all data underlying the findings in their manuscript fully available?

Reviewer #1: Yes

Reviewer #2: Yes

5. Is the manuscript presented in an intelligible fashion and written in standard English?

Reviewer #1: Yes

Reviewer #2: Yes

6. Review Comments to the Author

Reviewer #1: All the changes were done and now the article is clear and suitable to be published. Authors reviewed the English language for mistakes and included references that were necessary.

Reviewer #2: (No Response)

7. PLOS authors have the option to publish the peer review history of their article (what does this mean?). If published, this will include your full peer review and any attached files.

Reviewer #1: Yes: raquel braz assuncao botelho

Reviewer #2: Yes: Narender Raju Panjagari

---

## [Editor Report · Acceptance letter]

7 May 2020

PONE-D-19-34660R1 

Southern-style *Pad Thai* sauce: from traditional culinary treat to convenience food in retortable pouches 

Dear Dr. Chaijan:

I am pleased to inform you that your manuscript has been deemed suitable for publication in PLOS ONE. Congratulations! Your manuscript is now with our production department. 

With kind regards,

on behalf of

Dr. Edy de Brito 

Academic Editor

PLOS ONE